# Determinants of the Management of Native Vegetation on Farms

Geoff Kaine [1,*] and Vic Wright [2]

1   Manaaki Whenua—Landcare Research, Hamilton 3216, New Zealand
2   UNE Business School, University of New England, Armidale, NSW 2351, Australia; vwright5@une.edu.au
*   Correspondence: kaineg@landcareresearch.co.nz

**Abstract:** The clearing of native vegetation on private agricultural land has contributed greatly to the loss of ecosystems and biodiversity worldwide. Native vegetation on private land may be cleared for a variety of reasons, of which the expansion of agriculture is only one. In this study, we investigate how the clearing of native vegetation on private land is influenced by (1) the utilitarian, social and hedonic objectives of landholders and (2) the way in which the presence of native vegetation interacts with the farm system to contribute to, or detract from, achieving those objectives. Using data from a survey of agricultural landholders in New South Wales, Australia, we found that the landholders' management of native vegetation was strongly influenced by their perceptions of the opportunities and threats the native vegetation on their properties presented to them. The implications are drawn for predicting the clearing of native vegetation and designing effective policy interventions to influence the extent of clearing.

**Keywords:** native vegetation; motivation; land clearing; biodiversity policy; agricultural policy; Australia

## 1. Introduction

The clearing of native vegetation on private agricultural land has contributed greatly to the loss of ecosystems and biodiversity worldwide. In many countries, central and provincial governments have sought to preserve vulnerable species and habitats by placing regulatory controls on the clearing of native vegetation on private land, as has been the case in New South Wales, Australia [1–3].

The literature on native vegetation has tended to focus either on the planting or retention of native vegetation [4–7] or the clear-felling of forests to establish plantation or livestock agriculture [7]. The actual clearing of native vegetation, and the reasons for it, on private land in regions that were extensively cleared in the past, has received less attention even though continued clearing may further endanger vulnerable species and habitats. Furthermore, native vegetation may be cleared for a variety of reasons, of which the expansion of agriculture is only one. In this study, we investigate the impact of farm context on the management and clearing of native vegetation on private land in New South Wales, Australia, and consider the implications for future clearing and relevant public policy.

## 2. Background

Native vegetation in Australia may be managed in a variety of ways. For example, vegetation may be selectively cleared for the following reasons:

- For the construction of rural infrastructure like sheds, outhouses and powerlines;
- To prevent personal injury or property damage, including firebreaks;
- For construction timber or firewood.

Native vegetation may be clear-felled to expand livestock grazing and other agricultural activities like cropping. In addition, landholders may plant native vegetation and engage in environmental protection works such as bush regeneration, wetland protection, erosion protection, ecological burning and controlling weeds.

The clearing of native vegetation in New South Wales has been subject to regulation in a variety of ways since 1881 [8]. Comprehensive legislation to specifically regulate the clearing of native vegetation was introduced with the Native Vegetation Conservation Act in 1997, which was subsequently replaced by the Native Vegetation Act in 2003. This Act has been amended several times following reviews in 2009 and again in 2013 [8]. The clearing of native vegetation is now regulated under the Local Land Services Act 2013 and the Biodiversity Conservation Act 2016 [8]. The Office of Environment and Heritage, in partnership with Local Land Services (LLS), the Biodiversity Conservation Trust and the Department of Planning and Environment, is responsible for administering various components of the legislation, but Local Land Services is the primary regulatory authority on a day-to-day basis [9]. Under these Acts, it is an offence to clear native vegetation in contravention of the legislation. Clearing is permitted without approval on rural land depending on its category under the Act. Clearing without approval is also permitted for 'allowable activities' such as reducing an imminent risk of personal injury or damage to property, for sustainable grazing and for building and operating rural infrastructure, such as fence lines, dams, sheds and tracks [9].

As part of a statutory review of relevant sections of the Local Land Services Act 2013, LLS took the opportunity to survey landholders across NSW about their management, including clearing, of native vegetation. The authors, among others, advised LLS on the design of the survey. The survey sought information from respondents about their perceptions of the native vegetation on their property, their management activities and their management objectives. This information provided the basis for our investigation of the role of farm context [10–15] in respondents' management of native vegetation.

## 3. Theory

Crouch [10] demonstrated how the correlations between a range of practices for managing pastures, livestock breeding and livestock health in sheep enterprises reflect the logical interrelationships and practical interdependence between these practices. Relatively low correlations were found between practices that generate benefits independently of each other such as spring lambing and inoculation of clover seed. On the other hand, high correlations were found between practices that generate benefits jointly such as fodder conservation, weaner nutrition, and mulesing and weaning ages, all of which are practices that directly contribute to maximising lamb survival. Crouch [10] used these examples to illustrate how the benefits to be had from introducing a new practice or technology into a farming system depend on the way the interrelationships between the components in the system are modified and how these modifications contribute to, or detract from, achievement of the objectives of the primary producer.

Crouch [10] went on to argue that the process of farm development can be characterised as the ordered adoption of a succession of practices and technologies. In the early stages of farm development, these can be adopted relatively independently of each other. However, as development proceeds, the adoption of newer practices and technologies comes to depend increasingly on the prior adoption of older practices and technologies. A stage in farm development is reached where, in a practical sense, the random selection of new practices and technologies cannot continue [10,16–18]. Further development requires integrating new practices and techniques with the complex mix of technologies and practices adopted earlier. At this stage, the selection of practices is neither random nor haphazard; it is path-dependent.

The concept of farm context has been employed to explain farmers' behaviour in relation to planting trees [19], fencing streambanks [11] and the adoption of a variety of farm practices and agricultural technologies [13,17–20].

These considerations suggest that the adoption of farm practices requires the identification of those components and relationships within a farm system that are functionally related to the practice since they influence the benefits to be had from the practice. For ease of expression, we will term these components and relationships the farm context for

the practice. In other words, the farm context for an agricultural practice is defined by elements in the farm system that are functionally related to it (such as resources, constraints, agricultural technology and management practices, and strategies for managing risks) and so influence the achievement of the utilitarian, social and hedonic objectives of the farmer.

In the context we are considering here, this means that farmers' management of native vegetation on their properties, including decisions to clear it, will depend on (1) the utilitarian, social and hedonic objectives of the farmer and (2) the way in which the presence of native vegetation interacts with the farm system to contribute to, or detract from, achieving those objectives. The implication is that farmers' decisions about the management of native vegetation on their properties are neither random nor haphazard but are influenced by the following:

- Their perceptions of the products and services (if any) generated by the native vegetation on their property (farm context). For example, native vegetation might be a source of timber or offer recreational opportunities.
- Their perceptions of the opportunities and threats the native vegetation on their property presents to the achievement of their objectives (farm context). For example, native vegetation may contribute to productivity by preventing erosion or detract from productivity by harbouring weeds and pests.
- Their perceptions of the contribution of native vegetation to the well-being of the community and future generations (aspirations). For example, conserving native vegetation on their property may contribute to the well-being of the community.
- The relative emphasis the farmer places on generating an income from native vegetation versus the motivation to protect native vegetation (farm objectives). For example, an emphasis on generating income may encourage clearing to expand agricultural operations.
- Complementary management activity (farm context). For example, the clearing of land to expand commercial agricultural activity might trigger selective clearing for infrastructure, timber production or improving safety. Furthermore, the clearing of land to expand commercial agricultural activity and selective clearing might trigger the planting of replacement native vegetation.
- The nature of their agricultural enterprise (farm context). For example, livestock enterprises may require clearing vegetation for grazing that might otherwise be retained as a wind buffer for a horticultural enterprise.

Consequently, we hypothesised that the actions taken by farmers in managing the native vegetation on their properties would depend on their motivation to protect native vegetation; the emphasis they place on generating an income from their native vegetation; and their perceptions of its characteristics, the products and services it supplies and the effort required to manage it. All this is linked to the farm context, which includes their perceptions and preferences.

It follows, then, that respondents contact LLS regarding native vegetation, and the purpose of that contact would be influenced by their motivation to protect native vegetation, the activities they carried out in managing their native vegetation, their perception of the importance of native vegetation as a community asset and their views on the importance of experience and expert advice in making management decisions about native vegetation. Alone, attitudes and perceptions related to native vegetation are not predictive of farmer behaviour; they are inevitably conditioned by the physical aspects of the farm context.

## 4. Materials and Methods

The NSW Department of Local Land Services (LLS) commissioned a commercial market research company to conduct a survey of rural landholders in NSW. The survey was designed by subject experts from the Department with some contributions from the authors; it combined telephone interviewing and an online questionnaire. Included in the survey was a series of questions concerning respondents' perceptions of the products and services generated by their native vegetation, their perceptions of the opportunities and threats created by their native vegetation, their views on managing native vegetation and

the management actions they had taken over the past five years in managing their native vegetation (see Supplement A). Respondents answered questions about their perceptions of, and views about, native vegetation using a five-point rating indicating their agreement or otherwise with a series of statements about native vegetation and its management. Information was also collected on landholders' enterprises and regional location. The survey was carried out in April and May of 2023 and was completed by approximately 2000 respondents, of which 1750 were suitable for our analysis.

We used involvement, a concept from the fields of social psychology and marketing used widely to measure people's interest in products and services [21–25], to measure the strength of respondents' motivation to protect native vegetation. Higher involvement is associated with stronger, stable attitudes and a higher propensity to act, while lower involvement is associated with weak, unstable attitudes and a lower propensity to act. It is perhaps important to observe here that, in principle, while respondents' perceptions of the characteristics of their native vegetation might influence their involvement with the notion of protecting it, the reverse is not necessarily the case. Involvement creates the motivational context, the engagement with the issue, in which information (including one's own perceptions) is, or is not, sought and used to refine attitudes.

Respondents' involvement with protecting native vegetation was obtained using a scale consisting of ten statements about protecting native vegetation based on the work of Laurent and Kapferer [26]. Respondents indicated their agreement with the statements in the scale using a five-point rating. Respondents' agreement ratings were averaged across the ten statements to calculate their involvement score, with a higher score indicating greater involvement.

Prior to testing our hypotheses, we employed factor analysis to simplify the process of identifying statistically significant influences on the management practices of respondents and avoid potential problems with multi-collinearity. The factor analysis generated composite, uncorrelated variables describing respondents' perceptions of the products and services generated by their native vegetation, their perceptions of the opportunities and threats created by their native vegetation, their perception of native vegetation as a community asset, their views on managing native vegetation in terms of effort and the importance of managing native vegetation to generate an income, and their opinion on the relative importance of property experience and expert advice in managing native vegetation.

The principle that respondents' perceptions of the characteristics of their native vegetation influenced their involvement with protecting native vegetation was evaluated by examining the correlations between their perceptions of the products, services, opportunities and threats created by their native vegetation and their involvement with protecting native vegetation.

To test our hypotheses about the management of native vegetation, we formed the dependent variable by aggregating respondents' reported clearing activity over the past five years into three categories:

- Clear-felling to increase grazing or to expand agricultural operations;
- Selective clearing for timber, infrastructure or reducing the risk of injury or damage;
- Planting to protect native vegetation or environmental works.

Our hypotheses regarding these management actions were tested using binomial logistic regression [27]. The explanatory variables were the composite variables from the factor analyses, together with dummy variables representing respondents' enterprises and the variable measuring their involvement with protecting native vegetation.

Our hypotheses regarding respondents' contact with LLS were also tested using binomial logistic regression [27]. The explanatory variables were involvement with protecting native vegetation, the three management activities, their perception of the importance of native vegetation as a community asset, and their views on the importance of experience and expert advice in making management decisions about native vegetation.

Two additional management variables, which were weakly correlated with the other management variables, were included in the regressions. These variables represented a

belief that native vegetation should be allowed to grow naturally (that is, not be managed) and that management decisions were the sole preserve of the respondent. Regional dummy variables were included in the regression for contact with LLS to apply for grants as the nature of grants varies across the state [28]. The regressions were estimated using the backwards (Wald) procedure in SPSS [29] to eliminate insignificant variables.

## 5. Results

### 5.1. Factor Analyses

The results of the factor analysis of respondents' perceptions about the functions performed by the native vegetation on their properties were summarised by two factors representing approximately 55% of the variance in the data (see Table 1). One factor represented services (ecosystem, cultural, recreational and aesthetic) provided by native vegetation while the other factor represented products (pasture, timber) that could be harvested from native vegetation.

**Table 1.** Factors identified for native vegetation as a source of products and services.

| Product or Service | Source of Services | Source of Products |
|---|---|---|
| Is important for the natural scenery and aesthetic qualities | 0.81 | |
| Helps protect cultural heritage | 0.80 | |
| Protects and helps manage environmental aspects such as water quality, soil conservation, native plants and animals | 0.79 | |
| Is important for shade or shelter | 0.67 | |
| Is important for recreational activities (e.g., camping, picnics, bike riding, horse riding) | 0.62 | |
| Provides an economic return from activities other than timber and grazing, such as biodiversity offsets, carbon credits | 0.55 | |
| Provides an economic return from timber and/or grazing | | 0.84 |
| Is important for stock grazing | | 0.75 |
| Is an important source of timber for my own use (e.g., firewood, property infrastructure) | | 0.64 |

Note: Values are Pearson correlation coefficients. Correlations < 0.30 omitted.

Respondents' perceptions about the characteristics of the native vegetation on their properties were also summarised by two factors representing approximately 68% of the variance in the data (see Table 2). One factor represented characteristics that are potentially damaging or harmful threats (harbouring weeds and pests, fire hazard) while the other factor represented potentially useful services or opportunities (controlling erosion, protecting water quality, conserving native plants and animals).

**Table 2.** Factors identified for characteristics of native vegetation.

| Characteristic | Threats | Opportunities |
|---|---|---|
| My native vegetation shelters feral animals | 0.83 | |
| My native vegetation harbours native pest animals | 0.81 | |
| My native vegetation is a harbour for weeds | 0.76 | |
| My native vegetation is a fire hazard | 0.67 | |
| My native vegetation is important to control erosion and protect water quality | | 0.88 |
| My native vegetation is important for the conservation of native plants and animals | | 0.87 |

Note: Values are Pearson correlation coefficients. Correlations < 0.30 omitted.

Respondents' perceptions about managing native vegetation were summarised by two factors representing approximately 65% of the variance in the data (see Table 3). One factor represented respondents' perceptions of the effort (the cost and time) required to manage their native vegetation while the second factor represented respondents' views on the degree to which native vegetation should be managed to generate an income.

**Table 3.** Factors identified for productive value and management.

| Management Characteristics | Management Effort | Managed for Income |
|---|---|---|
| My native vegetation requires active management | 0.72 | |
| My native vegetation is costly to manage | 0.82 | |
| Managing my native vegetation takes too much time | 0.79 | |
| My native vegetation should be managed to produce timber products (e.g., sawlogs, firewood, fence posts) | | 0.83 |
| My native vegetation should be used to contribute as much as possible to income from my property | | 0.80 |

Note: Values are Pearson correlation coefficients. Correlations < 0.30 omitted.

Respondents' perceptions about the importance of experience and expert advice in managing native vegetation were also summarised by two factors representing approximately 62% of the variance in the data (see Table 4). One factor represented respondents' view that their experience on the property and their own capacity to assess native vegetation meant that they were the best people to make management decisions. The other factor represented the respondents' view that expert advice and information were needed to manage the native vegetation on their property. Respondents' perceptions about the importance of the community and the future in managing native vegetation were summarised by a single factor representing approximately 72% of the variance in the data (see Table 5).

**Table 4.** Factors identified for experience and advice.

| Characteristic | Property Experience | Expert Advice |
|---|---|---|
| My experience on my property makes me the best person to make decisions about managing my native vegetation | 0.81 | |
| You need to be a qualified ecologist to know all about the native species on my property | −0.46 | 0.43 |
| I'm capable of assessing native vegetation on my property | 0.85 | |
| I seek out information to better understand and manage the native vegetation on my property | | 0.80 |
| I rely on Local Land Services or other experts to identify and provide advice about the native vegetation on my property | | 0.77 |

Note: Values are Pearson correlation coefficients. Correlations < 0.30 omitted.

**Table 5.** Factors identified for the role of community.

| Role | Community Asset |
|---|---|
| Protecting native vegetation is important for maintaining the natural beauty or aesthetic qualities of my area | 0.88 |
| It's important to consider the community when making decisions about protecting the native vegetation on my property | −0.73 |
| Protecting native vegetation will be important for future generations of my family | 0.87 |
| Protecting native vegetation is important for the future of my community | 0.91 |

Note: Values are Pearson correlation coefficients. Correlations < 0.30 omitted.

*5.2. Perception of Vegetation Characteristics and Involvement*

The association between respondents' involvement with protecting native vegetation and their perceptions of the products, services, opportunities and threats their native vegetation creates is reported in Table 6. The results indicate that a substantial proportion of the variance in respondents' involvement with protecting native vegetation is explained by their perceptions of the importance of the environmental, cultural, recreational and aesthetic services it provides.

**Table 6.** Correlation between involvement and characteristics of native vegetation.

| Characteristic | Involvement with Protecting Native Vegetation |
|---|---|
| Source of services | 0.61 *** |
| Source of products | −0.02 |
| Opportunities | 0.65 *** |
| Threats | −0.21 *** |

Note: Values are Pearson correlation coefficients. *** $p < 0.001$.

Note that, if our reasoning was mistaken, meaning that involvement with protecting native vegetation did influence respondents' judgements about the characteristics of their native vegetation, then involvement should be negatively correlated with their perceptions of the products and threats generated by native vegetation. While, as expected, involvement was positively correlated with perceptions of the environmental, cultural, recreational and aesthetic services it provides, we found no causal relationship in the other direction: there was no correlation between involvement and perceptions of products and only a weak correlation between involvement and perceptions of threats (see Table 6).

*5.3. Management Activities*

We had hypothesised that the actions taken by respondents in managing the native vegetation on their properties would depend on their involvement with protecting native vegetation and their perceptions of its characteristics, the products and services it supplies, the effort required to manage it and their management objectives. We also hypothesised that their management actions would depend on the types of enterprise operated by respondents. We also expected that the clearing of land to expand commercial agricultural activity might trigger selective clearing for infrastructure, timber production or improving safety. And we expected that both the clearing of land to expand commercial agricultural activity and selective clearing might trigger the (replacement) planting of native vegetation. The results of the binomial regressions testing these hypotheses are reported in Table 7.

**Table 7.** Logistic regression estimates for management activities.

| | Clearing for Grazing and Agricultural Expansion | Selective Clearing | Planting and Environmental Works |
|---|---|---|---|
| Involvement with protecting native vegetation | | | 1.381 ** |
| Native vegetation is a source of services | | | 1.185 * |
| Native vegetation is a source of products | 1.203 * | 1.446 *** | |
| Native vegetation is a source of threats | 1.767 *** | 1.323 *** | 0.734 *** |
| Native vegetation is a source of opportunities | 0.584 *** | | 1.432 *** |
| Native vegetation is a community asset | | 0.868 * | |
| Native vegetation requires intensive management | | | 1.263 *** |
| Native vegetation should be left to grow as nature intended | 0.762 *** | 0.836 *** | |
| Autonomy in management | | | 0.814 *** |
| Native vegetation should be managed for income | 1.263 ** | 1.287 *** | 0.840 ** |
| Property experience is important in managing native vegetation | | | 1.259 *** |
| Expert advice is important in managing native vegetation | 1.234 * | | 1.186 ** |
| Selective clearing of native vegetation | | | 1.585 ** |
| Clearing for grazing or agricultural expansion | | 3.366 *** | 1.508 *** |
| Cropping | 2.464 *** | | |
| Livestock | 2.059 *** | 0.538 *** | |
| Lifestyle and hobby farming | | 1.375 ** | |
| Intercept | 0.094 *** | 2.086 *** | 1.213 |
| Nagelkerke $R^2$ | 0.27 | 0.21 | 0.22 |
| F-Test | <0.001 | <0.001 | <0.001 |

Note: Values are likelihood ratios. * $p < 0.05$, ** $p < 0.01$, *** $p < 0.001$ based on Wald test [27].

The results indicate that clearing for grazing and for agricultural expansion was more likely if respondents perceived that the native vegetation on their property can be a source of products, has potentially damaging or harmful characteristics and should be managed to generate an income. Such clearing was more likely, of course, on cropping and livestock properties. Clearing for grazing and for agricultural expansion was less likely if respondents perceived the native vegetation on their property as offering potentially useful services or opportunities, such as controlling erosion and conserving native plants and animals, or that it should be left to grow as nature intended. Clearing was also more likely if respondents believed that obtaining expert advice is important in managing native vegetation.

As expected, clearing for grazing and for agricultural expansion was a trigger for selective clearing (see Table 7). Selective clearing was also more likely if respondents perceived the native vegetation on their property as a source of products, as having potentially damaging or harmful characteristics and should be managed to generate an income. Selective clearing was more likely if respondents were lifestyle or hobby farmers and less likely if respondents were livestock farmers. Selective clearing was less likely the more respondents perceived their native vegetation as being a community asset and that it should be left to grow as nature intended.

As hypothesised, selective clearing and clearing for grazing and agricultural expansion were triggers for planting native vegetation and for carrying out environmental works. Planting and environmental works were also influenced by respondents' involvement with protecting native vegetation and their perceptions about native vegetation as a source of services and opportunities, together with believing that the vegetation on their property required intensive management (see Table 7). Respondents were less likely to carry out planting and environmental works the more they perceived that native vegetation is a source of threats, that it should be managed to generate an income and that the management of the native vegetation on their property was entirely up to them.

*5.4. Contact with Local Land Services*

We had hypothesised that respondents' contact with LLS regarding native vegetation, and the purpose of that contact, would be influenced by their involvement with protecting native vegetation and the activities they carried out in managing their native vegetation. We also hypothesised that respondents' contact with LLS might also be influenced by their views on the role of their experience and professional advice in managing vegetation, autonomy in decision-making, whether native vegetation should be managed at all and the importance of native vegetation as a community asset. The results of the binomial regressions testing these hypotheses are reported in Tables 8 and 9.

The results indicate that, not surprisingly, contact with LLS to notify the department of clearing activity, or to apply for a clearing certificate, was strongly influenced by planning to clear land to expand agricultural operations. Contact for support in assessing native vegetation was associated with selective clearing and clearing for agricultural expansion, as was contact to obtain information on the rules regulating clearing. Contact to apply for grants was associated with planting vegetation and environmental works.

Respondents who sought expert advice to assist them in managing native vegetation were generally more likely to have contact with LLS, especially regarding support in assessing the native vegetation on their properties, applying for a certificate to clear vegetation and applying for grants. Respondents who believed native vegetation should be left to grow as nature intended were less likely than other respondents to have contact with LLS generally, as were those who wished to preserve their autonomy in making management decisions. The latter were also less likely to contact LLS to apply for grants.

**Table 8.** Logistic regression estimates for contact with LLS about managing native vegetation.

| | Contact with LLS | Contact for Information on Rules | Applying for Grants |
|---|---|---|---|
| Involvement with protecting native vegetation | 1.397 ** | | 1.477 ** |
| Planting and environmental works | 1.353 * | | 1.759 * |
| Selective clearing | | 2.722 *** | 0.670 * |
| Clearing for grazing and agricultural expansion | 2.012 *** | 3.160 *** | |
| Property experience | | | |
| Seek advice about native vegetation | 1.924 *** | | 1.372 ** |
| Community asset | 0.703 *** | 0.637 *** | |
| Passive management | 0.808 *** | | |
| Autonomy in management | 0.810 *** | | 0.794 ** |
| Central Tablelands | | | 2.111 * |
| Riverina | | | 2.686 ** |
| Intercept | 0.326 * | 0.164 *** | 0.182 * |
| Nagelkerke R$^2$ | 0.15 | 0.24 | 0.18 |
| F-Test | <0.001 | <0.001 | <0.001 |

Note: Values are likelihood ratios. * $p < 0.05$, ** $p < 0.01$, *** $p < 0.001$ based on Wald test [27].

**Table 9.** Logistic regression estimates for contact with LLS about managing native vegetation (continued).

| | For Support in Assessing My Property's Native Vegetation Prior to Clearing | To Notify Local Land Services about a Clearing Activity | To Apply for a Certificate for a Clearing Activity |
|---|---|---|---|
| Involvement with protecting native vegetation | | | |
| Planting and environmental works | | | 0.349 *** |
| Selective clearing | 1.817 ** | | |
| Clearing for grazing and agricultural expansion | 3.352 *** | 4.427 *** | 9.286 *** |
| Property experience | | | 1.704 ** |
| Seek advice about native vegetation | 1.401 ** | | 1.651 ** |
| Community asset | | | 0.667 ** |
| Passive management | | | |
| Autonomy in management | | | |
| Intercept | 0.040 ** | 0.047 *** | 0.075 *** |
| Nagelkerke R$^2$ | 0.12 | 0.08 | 0.36 |
| F-Test | <0.001 | <0.001 | <0.001 |

Note: Values are likelihood ratios. ** $p < 0.01$, *** $p < 0.001$ based on Wald test [27].

## 6. Discussion

The results confirm that the landholders' management of native vegetation is strongly influenced by their perceptions of the opportunities and threats the native vegetation on their properties presents to them [30]. Clearing for the expansion of commercial agricultural activity was more likely if landholders perceived that the native vegetation on their property offered little in the way of opportunities for preventing erosion or conserving native plants and animals and, by harbouring pests and weeds, constituted a threat to the farm enterprise. Clearing for the expansion of commercial agricultural activity was also more likely if landholders thought managing native vegetation took too much time, money and effort and that native vegetation should generate an income.

These results suggest that broad-scale clearing for expanding grazing and other commercial agricultural activities was most likely when respondents perceived the native vegetation on their properties as offering them little commercial or conservation value. This result has some important implications for policies to conserve native vegetation. The first is that landowners are less likely to clear native vegetation for agricultural expansion if they perceive that there is the possibility of generating an income from it through passive mechanisms such as tax offsets, biodiversity credits or carbon credits [31–34]. Second, landowners may be less likely to clear native vegetation for agricultural expansion if they

can be compensated for the time and effort required to manage harboured pests and weeds that constitute a threat to productivity [32–36].

Third, confirming the accuracy of landowners' perceptions of the conservation value of native vegetation becomes important in safeguarding vulnerable species and habitats [32,36,37]. Respondents who had cleared land to expand their agricultural activities were more likely than other respondents to have contacted LLS to obtain information on the rules permitting the clearing of native vegetation and to seek support in assessing the native vegetation on their property prior to clearing. These results signal that LLS can influence the clearing of native vegetation to expand agricultural activities by providing advice and assessment services to landowners. We found that respondents who cleared to expand their agricultural activities were more likely, on average, to believe that having expert advice was important in managing native vegetation.

Selective clearing of native vegetation by respondents for timber, for firewood, to install infrastructure or to improve safety was more likely if respondents believed that the native vegetation on their properties was a source of products such as timber and grazing, should be managed to generate an income and was a source of pests and weeds. Selective clearing was also triggered by clearing to expand agricultural activities. These results suggest that landowners are less likely to clear native vegetation selectively if they believe that there is the possibility of passively generating an income from it (e.g., biodiversity credits, carbon credits) and if they can be compensated for the time and effort required to manage pests and weeds. However, the potential to influence these landowners is probably constrained either by the fact that clearing is necessary for safety reasons or to prevent damage to property, or that they would face additional costs if they were prevented from clearing (in terms of obtaining timber or having to relocate planned infrastructure).

Respondents who were selectively clearing were likely to contact LLS for information about rules permitting clearing and for advice on, or assessment of, the native vegetation on their property. They were, however, less likely to notify LLS of clearing activity than respondents who were clearing for agricultural expansion. These respondents may have believed that selective clearing was less likely to constitute a threat to vulnerable species or habitats compared to clearing to expand agricultural operations, and so felt there was no need for confirmation of their assessment that the native vegetation on their properties had limited conservation value. Selective clearing was less likely if respondents viewed their native vegetation as a community asset. These respondents were also less likely to have contact with LLS, possibly because they were less likely to be selectively clearing.

The planting and protection of native vegetation by landholders was influenced by the strength of their motivation to protect native vegetation and perceiving native vegetation as a supplier of aesthetic, cultural, recreational and ecosystem services. Planting was more likely if landholders perceived their native vegetation as offering opportunities and needing committed management. Planting was less likely if landholders perceived native vegetation on their property as a haven for pests and weeds and that it should be generating an income. Planting and environmental works were also triggered by selective clearing or clearing for agricultural expansion. These respondents were more likely than others to contact LLS about applying for grants but less likely than others to contact LLS for an assessment of their native vegetation. These respondents believed that the native vegetation on their properties was valuable from a conservation perspective. Confirming this to be the case may be worthwhile in ensuring that grant monies are allocated effectively.

Not surprisingly, respondents who believed that native vegetation should be allowed to grow as nature intended were less likely than other respondents to clear land or to have contact with LLS. The low rate of contact with LLS by these respondents may mean they may not have a full appreciation of the conservation value of their native vegetation and how best to manage it. Respondents who valued autonomy in decision-making were also less likely than other respondents to have contact with LLS. The latter result is concerning as these respondents may manage their native vegetation inappropriately, or even choose to clear it, without a full appreciation of its conservation value.

We found that respondents' motivation to protect native vegetation did not influence their propensity to clear land for agricultural expansion nor to engage in selective clearing, though it did influence their propensity to plant vegetation and engage in environmental protection. This result suggests that, while respondents may aspire to protect native vegetation, such aspirations are a subordinate consideration to economic considerations. This means respondents are unlikely to enact their aspirations to protect native vegetation if those aspirations conflict with safeguarding the business performance of the farm [3,31]. This reinforces the importance of developing policy interventions, to reduce the rate of clearing, that enable landholders to generate income from native vegetation. It also means that calls for campaigns to promote a stronger environmental ethic among farmers are unlikely to significantly reduce clearing [38,39], even if they were successful in increasing farmers' motivation to protect native vegetation. Such campaigns might, however, prompt an increase in the planting of native vegetation and environmental works [30].

It is perhaps important to observe again that, in principle, while respondents' perceptions of the characteristics of their native vegetation might influence their involvement with protecting native vegetation, the reverse is not the case. The foundation of involvement with an issue (protecting native vegetation) is the degree to which that issue affects the achievement of functional, experiential and self-expressive needs. If respondents believed the native vegetation on their property was a harbour for pests and weeds and had little or no conservation or productive value, then respondents are likely to believe that the native vegetation on their property contributes little to meeting their needs. Consequently, irrespective of their involvement with protecting native vegetation generally, they are unlikely to be strongly motivated to conserve the vegetation on their property. This leads to the conclusion that, rather than devoting public resources to trying to change farmers' interest in, or attitudes towards, protecting native vegetation, public resources should be devoted to ensuring that farmers' judgements about the conservation value of their native vegetation are accurate [36,37].

## 7. Conclusions

The clearing of native vegetation on private agricultural land has contributed greatly to the loss of ecosystems and biodiversity worldwide. A review of legislation governing the clearing of native vegetation in New South Wales, Australia, provided us with the opportunity to investigate the reasons why landholders, particularly farmers, had cleared native vegetation. Our results confirmed that the landholders' management of native vegetation is strongly influenced by their perceptions of the opportunities and threats the native vegetation on their properties presents to them.

Confirming the accuracy of landowners' perceptions of the conservation value of native vegetation becomes important in safeguarding vulnerable species and habitats. We found that respondents who had cleared land to expand their agricultural activities were more likely than other respondents to have contacted LLS to obtain information on the rules permitting the clearing of native vegetation and to seek support in assessing the native vegetation on their property prior to clearing. These results indicate that LLS is in a position to influence the clearing of native vegetation by providing advice and assessment services to landowners.

**Supplementary Materials:** The following supporting information can be downloaded at: https://www.mdpi.com/article/10.3390/conservation4020012/s1, Supplement A: Landholder questionnaire.

**Author Contributions:** Conceptualisation, Methodology, G.K. and V.W.; Data Curation, G.K.; Formal Analysis, Writing—Original Draft Preparation, Writing—Review and Editing, G.K. and V.W.; Project Administration, G.K.; Funding Acquisition, V.W. and G.K. All authors have read and agreed to the published version of the manuscript.

**Funding:** This research was funded by Local Land Services, New South Wales. The drafting of the manuscript was funded by the New Zealand Ministry for Business, Innovation and Employment (Grant # C09X2103).

**Data Availability Statement:** The dataset is commercially sensitive but may be made available on request to NSW LLS.

**Acknowledgments:** We would like to express our thanks to Louise Askew (NSW NRC) and Liam Hogg (NSW LLS) for their assistance. Thanks also to our survey respondents.

**Conflicts of Interest:** The authors have no relevant financial or non-financial interests to disclose.

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
