# Peer review of "Determinants of the Management of Native Vegetation on Farms"

_conservation, doi:10.3390/conservation4020012_

Round 1
Reviewer 1 Report
Comments and Suggestions for Authors
Very good description of the problem how to manage native vegetation in relation to required incomes from farming and cattle keeping.
Author Response
No response was required by Reviewer 1.
Reviewer 2 Report
Comments and Suggestions for Authors
The title of the manuscript is not in line with the content, which discusses farmers' perceptions of native vegetation and its clearing.
Did all farmers interviewed have natural vegetation on their property? If not, how did this affect the survey results? Could native vegetation management actions depend on the type of vegetation and its area?
The description of the statistical analysis is rather poor. Some tests are not described at all.
How can the authors explain the fact that selective cutting or clearing for agricultural development triggers planting and environmental management? Does the fact that these respondents were more likely than others to have applied to the LLS for grants mean that they didn‘t received them and then cleared native vegetation?
The discussion needs to be rewritten, it does not differ much from the results, and some of the information is repeated. This section may be combined with Results.
The conclusions are written as a summary of the discussion, even repeating sentences in full.
Throughout the manuscript, there is a considerable amount of information that is not the result of research, but the references are not given.
Self-citation presented in groups, e.g. [7-12] and [8-12] does not seem acceptable.
L 73-89 too much of Crouch [7]
L 152!
Factor analyses is not result, but a tool for investigating variable relationships.
L 219 such a reference to tables is unacceptable.
The titles of all tables should be changed to more comprehensible, more detailed ones, so that their content can be understood without reading the text.
The content of Table 6 is unclear: correlations between Threats and Threats = -0.21?
L 375 Citation!
L 538 Needs to be corrected.
Reviewer 3 Report
Comments and Suggestions for Authors
Natural environment disappearance is one of the key three threats that we face now, and native vegetation is an important natural element which has lots of ecological and cultural functions. The manuscript discussed the native vegetation management in NSW based on online questionnaire and telephone interviewing, and some interesting results and conclusions were obtained accordingly. While the manuscript needs some modifications before it can be considered for publication. The length should be reduced because there are some repetitive illustrations in most sections and also throughout the whole manuscript. Some specifics comments are as following:
l Title: which is too broad, and not corresponding to the main contents or research target of the manuscript, and needs modifications.
l Abstract: the main results were not fully illustrate and the last sentence is too long; the final conclusion is not specific, and the key methods and points were not fully described.
l Introduction: more information needs about native vegetation clearing and its effect, and also the clear target for this manuscript; suggest combine introduction, background and theory section as Introduction, and reduce its length of illustrations.
l Materials and methods: more explanations needed about representativeness of the questions that the questionnaire cover, and the statistic of obtained results is simple.
l Results: the results can be demonstrated using lines or figures, or pie chart, instead of firsthand results of the questionnaire, and table 6 can be deleted.
l Discussion: the current descriptions in the section mostly focus on literature summary while should focus on the questionnaires results and related explanations, and also the limitations that this study has.
l Conclusion: suggest to give some advice on how manage the native vegetation clearing activities of landholders, and also the relations between landholders and LLS, based on your research and questionnaire.
Comments on the Quality of English LanguageNo advice
Reviewer 4 Report
Comments and Suggestions for Authors
While the manuscript demonstrates overall quality, following suggestions are recommended for consideration.
i. Double check the punctuation in lines 87 and consider revising the sentence structure in line152.
ii. Consider revising the lines 157 to 160 and lines 190 to 194 to enhance readability. The line 190 to 194 can be improved or consider breaking it into separate sentence.
iii. In subsection 5.1 of “Results”, there are percentage values given in a sentence. Please clarify how it was obtained.
iv. Consider revising lines 322 to 323 for more clarity.
v. Please ensure the citation format is same throughout the manuscript. Double-check the citation format in line 375.
vi. The citation is kept in square brackets in the manuscript. Change the citation format from parenthesis to square brackets in line 397.
vii. Certain sentences are repeated. To avoid the redundancy, revisit lines 410 to 413 of “Result” and lines 503 to 507 of “Conclusion”.
viii. Lines 426 to 428 and lines 449 to 451 are repeated in “Result”. Consider paraphrasing it.
ix. The references are not in line with the format of the journal. Revise the “References”.
Round 2
Reviewer 2 Report
Comments and Suggestions for Authors
References are not given L 25-26, L 53-54
Author Response
We have added references to support lines 25-26 and lines 53-54 as requested by Reviewer 2.